# Frequency and Management of Acute Poisoning Among Children Attending an Emergency Department in Saudi Arabia

**DOI:** 10.3390/pharmacy8040189

**Published:** 2020-10-14

**Authors:** Mansour Tobaiqy, Bandar A. Asiri, Ahmed H. Sholan, Yahya A. Alzahrani, Ayed A. Alkatheeri, Ahmed M. Mahha, Shamsia S. Alzahrani, Katie MacLure

**Affiliations:** 1Department of Pharmacology, College of Medicine, University of Jeddah, Jeddah 21512, Saudi Arabia; 2College of Applied Medical Sciences, University of Jeddah, Jeddah 21512, Saudi Arabia; ban87dar@gmail.com (B.A.A.); sholanx@gmail.com (A.H.S.); 3Inspection Department, Saudi Food and Drug Authority, Jeddah 21512, Saudi Arabia; 4Department of Pharmacy, East Jeddah Hospital, Jeddah 22253, Ministry of Health, Saudi Arabia; Chemist_007@hotmail.com (Y.A.A.); Ayed2022@hotmail.com (A.A.A.); 5Department of Pharmacology, College of Medicine, Umm al Qura University, Makkah 24381 8073, Saudi Arabia; 6Department of Emergency, East Jeddah Hospital, Ministry of Health, Jeddah 22253, Saudi Arabia; ammaha@moh.gov.sa (A.M.M.); shamsiaza@yahoo.com (S.S.A.); 7Independent Research Consultant, Aberdeen AB24, UK; katiemaclure@aol.com

**Keywords:** poisoning, pediatric poisoning, acute poisoning, unintentional poisoning, drug poisoning, chemical poisoning, emergency department

## Abstract

**Background**: Acute poisoning is one of the common medical emergencies in children that leads to morbidity and mortality. Medications and chemical agents play a major role in these adverse events resulting in social, economic, and health consequences. **Aims of the study**: This study aimed to evaluate the frequency and management of acute poisoning among children attending the emergency room at East Jeddah Hospital, Jeddah city, Saudi Arabia. **Methods:** This study was a retrospective chart review of all acute pediatric poisoning incidences in children (0–16 years of age) from October-21-2016 to March-03-2020 who were attending the emergency department. Data were analyzed via SPSS software. **Results:** A total of 69 incidences of acute poisoning in children who attended the emergency department at East Jeddah Hospital; males (n = 38, 55.1%). Most children were aged 5 years or younger (n = 41, 59.4%). Unintentional poisoning occurred among 56.5% of observed cases of which 52.2% occurred in children younger than 5 years; 7.20% (n = 5) of patients were 12 to 16 years of age and had deliberate self-poisoning. The association between type of poisoning and age groups was statistically significant (chi-square = 28.5057, p = 0.0001). Most incidences occurred at home (n = 64, 92.8%). Medicines were the most common cause of poisoning (n = 53, 76.8%). An excessive dose of prescribed medicine poisoning accidents was reported in 10.1% cases. Analgesics such as paracetamol were the most documented medication associated with poisoning (39.1%) followed by anticonvulsants and other central nervous system acting medicines (18.8%). The most common route of poisoning was oral ingestion (81.2%). One mortality case was documented. **Conclusion:** Although not common, accidental and deliberate acute poisoning in children does occur. More can be done to educate parents on safe storage of medicines, household cleaning and other products associated with acute poisoning in children. Likewise, children can be taught more about the risks of poisoning from an early age. As importantly, clinicians need to include more detailed notes in the electronic medical records (EMR) or the system needs to be improved to encourage completeness to more accurately inform the research evidence-base for future service design, health policy and strategy.

## 1. Introduction

Acute pediatric poisoning remains a worldwide health issue that requires medical attention at hospital emergency departments with consequences of morbidity and mortality [1,2,3,4]. It has social, economic, and health implications especially in children under the age of five who account for the largest percentage of poisonings globally [1,2,3,4,5,6]. The outcomes of poisoning range from mild incidences to severe complications or death, with most pediatric poisoning occurring accidentally by oral ingestion [1,2,3,4,5,6]. According to the World Health Organization (WHO) report (2008), an estimated 45,000 fatalities occur annually amongst children and young people (aged under 20) linked to acute poisoning [5]. In 2015, the American Association of Poison Control Center (AAPCC) reported that more than 1.3 million children were exposed to poisoning substances, 40% of whom were children less than 3 years old [6].

Studies by Berta et al. (2020) in Northern Italy [7], Patel et al. (2017) in the United States [8], Ahmed et al. (2015) in Qatar, [9], Mintegi et al. (2006) in Spain [10], Hoy et al. (1999), Reith et al. (2001), Schmertmann (2013) and Lee et al. (2019) in Australia [11,12,13,14], Mutlu et al. (2010) in Turkey [15], Azab et al. (2016) in Egypt [16], Sharif et al. (2003) in Ireland [17], Tobaiqy et al. (2010) in Scotland [18], Christophersen (2002) in Denmark [19], Dayasiri (2017 and 2020) in Sri Lanka [20,21], Mohammadi et al. (2020) in Iran [22], Yehya et al. (2020) and Albals et al. (2020) in Jordan [23,24] and Ham et al. (2020) in Korea [25] demonstrate the global nature and breadth of research of the problem of acute poisoning. The most frequent source of poisoning varies from country to country depending on social, economic, cultural, and educational background [1,2,3,4,5,6,7,8,9,10,11,12,13,14,15,16,17,18,19,20,21,22,23,24,25]. In developed countries, the most toxic substances are medicines and household cleaning products; developing countries more frequently see kerosene and pesticides causing acute poisoning in children [1,2,3,4,5,6,7,8,9,10,11,12,13,14,15,16,17,18,19,20,21,22,23,24,25].

Several studies of acute pediatric poisoning have been conducted in Saudi Arabia [26,27,28,29,30]. A review by Izuora and Adeoye (2001) in Hafr Al Batin of seven years of case notes of children admitted to a single military hospital in the Eastern Province identified 168 accidental pediatric poisonings out of 9951 pediatric admissions (1.7%). This was most common in children aged between 1 and 3 years old (63%). Most poisoning cases were related to medicines which accounted for 108 cases (64.3%) and household materials (n = 60, 35.7%) such as cleaning products [26].

A study by Al-Shehri (2004) in Aseer Province looked at patterns of poisoning in 114 children aged 12 years and under [27]. They concluded that the peak age for poisonings in children is before the age of four, mainly in summer with medical drugs the most common agents of poisoning. Further, they noted living rooms and bedrooms were the places where most incidents of poisoning occurred. Their recommendations were that “Good and continuous supervision by parents is essential, especially from the age 1–5 years”, and that, “There should also be legislation for the use of child resistant containers for home medicines and household agents” [27].

Abd-Elhaleem et al. (2014) also looked at patterns of acute poisoning but in both adults and children in Al Majmaah region [28]. Their retrospective review of 169 records from pediatrics concluded with a recommendation on, “implementation of legislations to ban over the counter selling of medications and to sell potentially dangerous chemicals in childproof containers” also, “Improving proper and complete medical record-keeping” [28].

Another retrospective study of all pediatric poisoning cases reported to the Drug and Poison Information Center in Riyadh, Saudi Arabia by Alghadeer et al. (2018) identified 735 children presented to the Pediatric Emergency Department with poisoning from January 2010 to December 2016. Children younger than two years of age (n = 459, 62%) were significantly affected by poisoning. Drug overdose (n = 119, 92.2%) was the major cause of poisoning and analgesics were the most commonly reported medicine (n = 26, 20.4%) [29]. They concluded with a recommendation, “to help to develop national plans to decrease the financial burden of emergency department congestion and hospital crowding” [29].

Most recently, and also based in Riyadh, Alruwaili et al. (2019) aimed to influence the design and implementation of effective preventative strategies to reduce pediatric poisoning nationally (30). Their review focused on 2 years of case notes of 1035 patients aged 12 years and younger. Their findings showed that, “Household products were the commonest reason for pediatric poisonings in Saudi Arabia and most of them were asymptomatic. Our results suggest a need for strategic plans for prevention and care” [30].

The current literature about poisoning among children in Saudi Arabia is limited and lacking in Jeddah in terms of examination of poisoning agent and the medications used for management and clinical outcomes [26,27,28,29,30].

## 2. Aims

This study aimed to determine the frequency and management of acute pediatric poisoning in East Jeddah Hospital in Jeddah city, Saudi Arabia. The objectives of the study were:To determine the number of incidences of accidental and deliberate acute poisoning.To identify the patterns of poisoning.To describe the sources and routes of poisoning.The describe the management of treatment.To describe the outcomes of cases of poisoning in children.

## 3. Methods

This study is a retrospective medical chart review of acute pediatric poisoning cases attending the emergency department of East Jeddah Hospital in Jeddah city, Saudi Arabia. Jeddah is the largest city in Makkah Province of Saudi Arabia with an estimated 3.5 million population (2020). East Jeddah Hospital is one of the largest of five Ministry of Health Hospitals in the city, which serve the public and community in the east of Jeddah with 300 beds capacity. In this study, all children 16 years or younger attending as a result of acute poisoning from 21 October 2016 to 3 March 2020 were included.

### 3.1. Data Collection and Statistical Analysis

The data were collected from patients’ electronic medical records (EMR) by two pediatric physicians and two pharmacists who work at the emergency department. The data collection sheet was comprised of the patient’s demographic profile such as age, gender, weight, nationality, patient medical history, vital signs, the place where the poisoning occurred, the type of poisoning, route of intoxication, symptoms of intoxication, the medical treatment, antidote given, and the outcome of medical management. Data analysis used SPSS (SPSS Inc., Cary, NC version 20.0) and comprised descriptive statistics including frequencies and Pearson’s chi-squared test which was used to determine any association between (i) causes of toxic exposures and (ii) child’s age group; *p* < 0.05 was considered statistically significant.

### 3.2. Ethical Approval

Management authorization was gained from the Saudi Arabia Ministry of Health (MOH) Reference Number (01190) and ethical approval from King Abdulaziz City for Science and Technology (KACST), KSA: (H-02-J-002).

## 4. Results

There were 69 incidences of acute poisoning recorded in children aged 16 and younger attending the emergency department.

### 4.1. Patterns of Poisoning

Demographic data are provided in Table 1. There were 38 boys (55.1%) and 31 girls (44.9%). Most were under 5 years of age (n = 41, 59.4%), and most were Saudi citizens (n = 61, 88.4%). As described in Table 2, more than half of the documented poisoning cases occurred unintentionally (n = 39, 56.5%), and were for children 5 years of age and younger (n = 36, 52.2%). The majority of these incidents occurred at home (n = 64, 92.8%). Intentional poisoning occurred in five cases (7.2%) all for children aged 12 to 16.

### 4.2. Sources and Routes of Poisoning

Table 3 includes data on the sources and routes of poisoning. Acute poisoning due to an excessive dose of a prescribed medicine was reported (n = 7, 10.1%). Around a quarter of the incidents (n = 18, 26.1%) were reported as unclassified with no further information on the EMR. The association between the pattern of poisoning and age groups was statistically significant (chi-square = 28.5057, p = 0.0001). The main source of acute poisoning was medicines (n = 53, 76.8), followed by intoxication by other chemical substances (n = 9, 13%) with the remainder due to unknown toxic materials (n = 7, 10.1%). Oral ingestion was found to be the most common route of poisoning (n = 56, 81.2%). Analgesics and antipyretics such as paracetamol were the most common pharmaceutical agents that caused poisoning (n = 27, 39.1%) followed by central nervous system (CNS) active medicines such as anticonvulsants/CNS acting medicines (n = 13, 18.8%), antipsychotics (n = 9, 13%), cardiovascular medicines (n = 3, 2.9%), and one incidence of poisoning by antihistamines (n = 1, 1.4%). Chemical poisoning (n = 9, 13%) was caused by heavy metals and organophosphate compounds. Seven incidents of poisoning (10.1%) occurred in which the exact type of chemical had not been identified.

### 4.3. Management of Poisoning 

Treatment of poisoning incidents varied from one case to another depending on patient condition, type of poisoning and time of exposure. In this study, treatment intervention was reported in most cases as shown in Table 3. Antidote pharmacological treatment was given to 31 patients (44.9%), supportive treatment in 15 (21.7%), however 23 (33.3%) required no medical intervention. Of those who received antidotes, (n = 16, 51.6%) were given activated charcoal, N-acetylcysteine (n = 9, 29%), antihistamines (n = 3, 9.7%). One patient received naloxone, atropine and fomepizole (Table 3).

### 4.4. Outcome of Poisoning

A total of 25 (36.2%) children were admitted to the pediatric ward, while 35 (50.7%) were discharged from the emergency department after receiving treatment and eight (11.6%) were admitted to the Pediatric Intensive Care Unit (PICU). One case of mortality was documented in PICU (n = 1, 1.4%).

## 5. Discussion

This study aimed to determine the frequency and management of acute pediatric poisoning in an emergency department which had been found to be lacking in the Saudi Arabian literature, particularly in Jeddah [26,27,28,29,30].

### 5.1. Frequency of Poisoning

There were 69 incidences of acute poisoning attending the emergency department at this single hospital in Jeddah over ~3.5 years. Most incidents were in boys which is consistent with other recent studies [27,28]. Although there have been no comprehensive epidemiological studies of the incidence of poisoning in the Kingdom of Saudi Arabia to date, studies published for different Gulf regions and time periods also indicate that children 4 or 5 years and younger are most often affected; similar results were reported in studies worldwide [1,7,9,11,14,21,22,23,24]. The literature notes that the balance switches in adolescents when girls were more likely to attend hospital emergency departments with intentional poisoning [1,15].

### 5.2. Patterns of Poisoning

In the literature, most incidences of poisoning occurred at home supporting our study’s finding of 92.8% [1,3,13,15,17,21,24,29]. Unintentional poisoning accounted for most incidents in this study (56.5%) while intentional self-poisoning was reported in five incidents. Intentional poisoning usually affected adolescents, in a study of 148 cases of poisoning, 86% were accidental; of the intentional cases, 33% were suicidal in subjects 12 years or older [17]. Of note, attempted suicide is a criminal act in Saudi Arabia; social stigmatization associated with psychiatric illness and their medications is another issue, both may play a role in under-reporting of attempted intentional poisoning. Unintentional poisoning is reported in the literature to be due to several reasons including child’s curiosity, natural tendency to explore the environment, and a lack of awareness of surrounding risk. As previously reported, parents and guardians must be responsible for the safe storage of medicines and household cleaning products while also educating children about potential risk of poisoning [9,16,17,20,27].

### 5.3. Sources and Routes of Poisoning

In this study, ingested medicines were found to be the main cause of acute poisoning (73.9%) consistent with previous studies [1,8,9,10,19,29] followed by chemical poisoning that occurred in 13% of the incidents. According to the Annual Report of AAPCC (2018) National Poison Data System (NPDS) [6], the top five most common exposures in children age 5 years or less were cosmetics and personal care products (12.1%) followed by household cleaning substances (10.7%) and analgesics (9.0%).

Analgesics—specifically non-steroidal anti-inflammatory drugs—as well as household cleaning substances are the most common poisons in children in recent studies [9,10,11,12,19,21,23,30]. In these studies, analgesics like paracetamol were the most documented cause of poisoning in children (39.1%) in addition to pharmaceutical products such as syrups that attract children due to colors and flavors followed by anticonvulsants (18.8%) and antipsychotics (13%) [1,28] Other studies have reported that neurological medicines were the most common drugs causing poisoning in children followed by analgesics [10,21].

### 5.4. Management of Poisoning

The management of poisoning in pediatric cases depends on several factors such as the type of poison, the dose, clinical manifestation, age, presence of other diseases or injury, and the time of poisoning exposure [4,5,6]. This study found that most poisoning cases in which there was antidote administration, were managed in the hospital through decontamination with activated charcoal (51.6%) or N-acetylcysteine (antidote of paracetamol; 29%). Previous studies have shown that activated charcoal alone is a better treatment for poisoning cases presenting in the hospital within one hour [19]. Activated charcoal can decrease absorption in the stomach and intestine for a wide variety of toxins and medicines such as carbamazepine, phenobarbital, theophylline, salicylates, and valproic acid [10,19,30].

### 5.5. Outcomes of Poisoning

In this study, 25 patients (36.2%) were admitted to the pediatric ward while (n = 8, 11.6%) were admitted to the PICU; in addition, one mortality was documented in PICU. The clinical severity of poisoning incidents in this study is higher than what is reported in similar research where most cases were mild with 101 (17.2%) cases admitted to the hospital and only 21 (3.6%) were admitted to the PICU [1].

Of note, most childhood poisoning incidences occurred due to the easy availability of medicines and chemicals at home in several forms and a lack of parental supervision to keep these materials in a safe place and out of the reach of children. Educating the community and particularly parents and guardians about the risks of drug and chemical poisoning in children may reduce the occurrence of such harmful adverse events [1,27,28].

### 5.6. Strengths and Limitations

All studies have unexpected limitations. The number of reported cases was low (n = 69) but that is a limitation of real-world data. Incomplete and missing data, specifically the uncategorised patients, may limit the generalizability of the study’s findings. However, when considered a learning point to recommend clinicians are more detailed and accurate in their EMR notes or that the system be improved to encourage completeness, this becomes a strength supported by previous literature in Saudi Arabia [28]. The results highlight the importance of community education and developing guidelines for monitoring and managing poisoning incidents in Saudi Arabia, which is again, supported by the existing literature in Saudi Arabia [27,29,30]. Further, it adds evidence from Jeddah to existing Saudi literature [26,27,28,29,30].

## 6. Conclusions

Although not common, unintentional and intentional acute poisoning in children does occur. More can be done to legislate and educate parents on safe storage of medicines, household cleaning and other products associated with acute poisoning in children. Likewise, children can be taught more about the risks of poisoning from an early age. As importantly, the number of unclassified patients in this study demonstrates that clinicians need to include detailed notes in the EMR or the system needs to be improved to encourage completeness, to more accurately inform the research evidence-base for future service design, health policy and strategy.

## Figures and Tables

**Table 1 pharmacy-08-00189-t001:** Demographic profile of children with poisoning (n = 69).

Profile	No. of Children	% of Children
**Gender**		
Male	38	55.1
Female	31	44.9
**Age group**		
0–5 yrs	41	59.4
6–11 yrs	18	26.1
12–16 yrs	10	14.5
**Nationality**		
Saudi	61	88.4
Non-Saudi	8	11.6
Total	69	100.0

**Table 2 pharmacy-08-00189-t002:** Patterns of poisoning incidences according to causes of toxic exposures in relation to age (n = 69).

Causes/Agen (%)	0–5 Years	6–11 Years	12–16 Years	Totals *
Unintentional Poisoning	36 (52.2)	3 (4.3)	0 (0)	39 (56.5)
Deliberate Self-Poisoning	0 (0)	0 (0)	5 (7.2)	5 (7.2)
Excessive Dose of a Prescribed Medicine	3 (4.3)	4 (5.8)	0 (0)	7 (10.1)
Unclassified(no details available in EMR)	3 (4.3)	11 (15.9)	4 (5.8)	18 (26.0)

* Please note there is a rounding discrepancy of 0.2.

**Table 3 pharmacy-08-00189-t003:** Summary of acute poisoning data.

n = 69, Unless Otherwise Stated	No. of Children	% of Children
**Site of Poisoning**		
At Home	64	92.8
At School	1	1.5
At Hospital	4	5.8
**Type of Poisoning**		
Medicines	53	76.8
Chemicals	9	13.0
Unknown	7	10.1
**Route of Intoxication**		
Oral	56	81.2
Local	4	5.8
Intravenous (IV)	2	2.9
Inhalation	1	1.5
Unknown	6	8.7
**Class of Medicine (n = 53)**		
Analgesics/Antipyretics	27	39.1
Anticonvulsants/CNS medicines	13	18.8
Antipsychotics	9	13.0
Cardiovascular medicines	3	2.9
Antihistamines	1	1.4
**Chemical Substances (n = 9)**		
Heavy Metals	3	4.3
Organic and hydrocarbon compound	3	4.3
Organophosphate	2	2.9
Alcohol/Methanol	1	1.4
Unknown (n = 7)	7	10.1
**Intervention**		
Antidote	31	44.9
Supportive Treatment	15	21.7
No need	23	33.3
**Antidote (n = 31)**		
Activated Charcoal	16	51.6
N-Acetylcysteine	9	29.0
Antihistamine	3	9.7
Naloxone	1	3.2
Atropine	1	3.2
Fomepizole	1	3.2
**Outcome of Poisoning**		
Discharged	35	50.7
Pediatric wards	25	36.2
Pediatric Intensive Care Unit (PICU)	8	11.6
Died in PICU	1	1.4

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
