# Peer review of "Frequency and Management of Acute Poisoning Among Children Attending an Emergency Department in Saudi Arabia"

_pharmacy, 2020, doi:10.3390/pharmacy8040189_

Round 1

Reviewer 1 Report

Introduction: Please explain statistical significance, which type of poisoning and which age group. The authors stated Chi-square and with that they need to also state the degrees of freedom.

Introduction: Please describe population of Eastern province and how many hospitals are there.

Why did you choose 16 years of age as cut of? Adolescent group is from 13 to 18 years of age (https://www.ncbi.nlm.nih.gov/mesh/68000293) and it is common way of grouping by age. And why groups 0-5, 6-11 and 12-16?

Methods: Authors should change statement "...comprised descriptive statistics." because you stated Chi-square results which is not descriptive statistics.

Results: Please explain big number of unclassified patients! Table 2. has sum of 99.9% not 100%.

Figure 1. Graph shows on y-axis decimal comma instead of decimal full stop. Rearrange the graph to see all text describing columns.

Explain abbreviations when you first state in text (i.e. IV).

Table 4. Authors skip class- cardiovascular medicines and it is missing 3 patients.

Figure 3. Replace decimal comma.

Outcome of poisoning: Please describe this one documented mortality. Did it happen on hospital admission or patient was treated in PICU? This mortality outcome should be new figure explaining that 68 patients were successfully treated, and 1 death.

Discussion: Authors should more explain and categorize Kingdom of Saudi Arabia by income when explaining WHO report.

Conclusion: The authors should also state education regarding the storing of medicines. Through whole document the authors used term medicines and in conclusion they use term drugs. Please correct.

Author Response

Many thanks,

Reviewer 2 Report

With a retrospective study, the Authors analyse substances that have caused intoxication in a population under 16. The topic is very interesting as the problem of poisoning among young people, both voluntary and involuntary, remains of great topicality. Nevertheless, the work requires some modifications

Abstract

  1. “Data” is plural
  2. The conclusion section does not report conclusion, but a summary of what has been just described in the results section

Introduction

Introduction is quite poor, perhaps expanding the horizon and considering realities also outside the Saudi Arabia could give more breath to the work, especially given that it is proposed to be published in a prestigious international journal.

  1. Line 44: authors cite the paper of Qazi and Saqib [2], which is centered in the in Jammu District (Jammu and Kashmir) India, to support their claim that intoxication “in children under the age of five [who] accounted for the largest percentage of poisonings globally”. I don’t think that the cited paper is an appropriate reference to this sentence
  2. Authors could draw global epidemiological data from more recent works, such as (as an example)
    1. Kyu HH, Stein CE, Boschi Pinto C, Rakovac I, et al. Causes of death among children aged 5-14 years in the WHO European Region: a systematic analysis for the Global Burden of Disease Study 2016. Lancet Child Adolesc Health2018;2:321-337.
    2. Mintegi S, Azkunaga B, Prego J, et al. International epidemiological differences in acute poisoning in Pediatric Emergency Departments. Pediatr Emerg Care2019;35:50-57.

Results

  1. Lines 101-102: “Acute poisoning due to excessive dose of a prescribed medicines was reported in (n =7, 10.1%) of the incidents”. Please correct the parenthesis (same typos in lines 139-141, 191…)
  2. Table 2. Please revise statistics and asterisks.
  3. Table 2 and figure 1 are redundant since they show exactly the same results (which are also described quite in detail in the text).
  4. Line 123: “73.91% (51) got poisoning by medicines followed by 13.04% (9). “ the sentence is incomplete
  5. Again, table 3 and figure 2 show the same data, this is unnecessary
  6. Line 138: “as shown in (Table 5).” Please remove parenthesis
  7. Table 5. All the data are already detailed in the text, the same for table 6
  8. Paragraph 4.5, table 7 and figure 3 repeat thrice the same data

Discussion

  1. Lines 154-155: is the difference between male and female patients significant? There are much more recent references than the one cited (10) to support the data reported by the authors
  2. Lines 172-174: what could be the explanation for the difference between the chemical substances involved in toxic exposure reported by the authors and the ones reported in the cited work?
  3. Line 176: I would hardly define “a recent work” the paper of Reith et al., published in 2001. There are more recent references, such as doi: 10.1186/s13052-020-00845-0. or doi: 10.1097/PCC.0000000000001187.
  4. Lines 185-186: I don’t agree with the sentence “Most poisoning cases were managed in the hospital through decontamination with active charcoal (51.6%), and N-acetylcysteine (antidote of paracetamol) was given to (29%) of the cases”, since active charcoal was given to 16 children/69 (23.2%), which is not “most poisoning cases”. It would be may be more appropriate to specify “Most poisoning cases in which there was antidote administration”

Conclusion

Conclusions are a summary of the results. Please rewrite proper conclusions, include the limits and strength of the study, and the value and significance/perspectives of the study

Author Response

Many thanks,
